# Kinetics of Eosinophils during Development of the Cellular Infiltrate Surrounding the Nurse Cell of *Trichinella spiralis* in Experimentally Infected Mice

**DOI:** 10.3390/pathogens10111382

**Published:** 2021-10-26

**Authors:** Vicente Vega-Sánchez, Fabián-Ricardo Gómez-De-Anda, Georgina Calderón-Domínguez, Mary-Carmen-del-Sol Ramírez-y-Ramírez, Nydia-E. Reyes-Rodríguez, Andrea-P. Zepeda-Velázquez, Raquel Tapia-Romero, Jorge-Luis de-la-Rosa-Arana

**Affiliations:** 1Instituto de Ciencias Agropecuarias, Universidad Autónoma del Estado de Hidalgo, Rancho Universitario, Tulancingo 43600, Mexico; vicente_vega11156@uaeh.edu.mx (V.V.-S.); nydia_reyes@uaeh.edu.mx (N.-E.R.-R.); 2Departamento de Graduados e Investigación de Alimentos, Escuela Nacional de Ciencias Biológicas, Instituto Politécnico Nacional, Prolongación de Carpio y Plan de Ayala, Ciudad de México 11340, Mexico; gcalderondominguez@gmail.com; 3Laboratorio de Inmunoparasitología, Instituto de Diagnóstico y Referencia Epidemiológicos, Secretaría de Salud, Francisco de P. Miranda 177, Unidad Lomas de Plateros, Ciudad de México 01480, Mexico; sol0182@hotmail.com (M.-C.-d.-S.R.-y.-R.); andrea_zepeda@uaeh.edu.mx (A.-P.Z.-V.); 4Laboratorio de Inmunología, Hospital Infantil de México, Secretaría de Salud, Dr. Márquez 162, Delegación Cuauhtémoc, Ciudad de México 06720, Mexico; raqtapia@yahoo.com.mx

**Keywords:** cellular infiltrate, eosinophil, nurse cell, experimental infection, *Trichinella spiralis*

## Abstract

We study the kinetics of eosinophils during the development of the cellular infiltrate surrounding the nurse cell of *Trichinella spiralis* (*T. spiralis*) in experimentally infected mice. Male CD1 mice were experimentally infected with 50 viable muscle larvae of the MSUS/MEX/91/CM-91 *T. spiralis* strain. Tongues and diaphragms were obtained daily from days 13 to 39 post infection. Diaphragms were compressed and subjected to Giemsa stain. Tongues were histologically sectioned and stained with erythrosine B or hematoxylin and eosin. The cellular infiltrate and the nurse cell-larva complex were detected by optical microscopy since day 16 post infection. The size of the larva increased exponentially during the course of the infection. The kinetics of eosinophils showed a multimodal trend, with a bimodal predominance. The maximum peaks were reached on days 21 and 27 post infection. The results of this study demonstrate that eosinophils occur abundantly in two transcendent moments of the *T. spiralis* life cycle: first, when the stage 1 larva invades the myocyte and second when the nurse cell-larva complex has been fully developed. These results help one to understand the immunobiology of *T. spiralis*, highlighting the importance of eosinophils in the survival of the larva in skeletal muscle. Further studies are needed to characterize the cell populations that comprise the cellular infiltrate during the development of the mother cell.

## 1. Introduction

*Trichinella spiralis* is a zoonotic parasite that infects a wide range of mammalian hosts. This nematode is unusual in two ways that have significance for the host immune response. First, the life cycle is completed in a single host. Second, the parasite resides in two distinct intracellular habitats, resulting in two distinct types of clinical manifestations: intestinal disorder (intestinal phase) and myopathy (muscular phase). The muscular phase begins with the infection of a myofibrilla with the stage 1 larva, which will develop to a muscular larva [1]. *T. spiralis* muscle larvae infect exclusively striated muscle cells, transforming them into “nurse cells”, where larvae stay for many years. The nurse cell quickly surrounds itself with a cellular infiltrate associated with antigens from the muscle larvae. Many authors have already reported the process of nurse cell formation [2,3,4,5], but there is scarce information about the characterization of the cell populations that constitute the cellular infiltrate. Although eosinophils are one of the main populations taking part in the helminth infection, the kinetics of development of the cellular infiltrate surrounding a nurse cell of *T. spiralis* are unknown.

*T. spiralis* newborn larvae penetrate the muscle cells by the 14th day of infection and continue to grow in size through days 20–30 post infection; thus, the nurse cell formation involves complex steps and events that take place over a 20-day period from the time of the initial larval invasion. In the beginning, the process is a response of host cells to larval invasion (dedifferentiation, cell cycle re-entry, and arrest), while in the later stage it is a reformation or a restructuration of host cell processes (activation, proliferation, and differentiation) induced by the larva; simultaneously, the nurse cell is surrounded by a thin capsule of connective tissue [4]. In an immunocompetent host, inflammatory infiltrates are formed around nurse cells where eosinophils are prominent but, in their absence, the muscle larvae die, which also correlates with enhanced levels of inducible nitric oxide synthase [6] and IFN-gamma [7]. Thus, eosinophils seem to act as immune response modulators avoiding the Th1 environment, which is detrimental to the survival of the muscle larva.

The staining properties of muscle larvae and nurse cell have been widely reported in hematoxylin and eosin-stained histological sections [8]. These properties are advantageous in the Giemsa stain to contrast the presence of *T. spiralis* in samples of experimentally infected mice [9] and rats [10]. Thus, the histologic studies of *T. spiralis*-infected muscle show nurse cells surrounded by infiltrates resembling a granuloma. Although the infiltrate must be composed of macrophages, eosinophils, scattered plasma cells, and lymphoblasts, the proportion of the sample that constitutes the infiltrate is controversial. In this regard, some studies have stated that the infiltrate is constituted by macrophages, while in others the presence of lymphocytes is reported as the main component [3,11]. Other studies have shown that in mice infected with non-encapsulated *T. pseudospiralis* or in experimentally infected nude mice, a markedly lower level of muscle inflammation is observed [8]. Coinfection with *T. pseudospiralis* and *T. spiralis* also leads to lower inflammation around *T. spiralis* infected cells [11]. Nurse cell capsule formation does not depend on the host immune response.

Since there are scarce and controversial studies about the characterization of the cellular infiltrate that surrounds the nurse cell, there are no conclusive data about the proportion of eosinophils during its formation. In this work, we documented this process in the murine model by using image analysis techniques. The purpose of image pre-processing was to obtain better image quality while keeping the original image dimensions. This computational method allowed us to reduce the noise in the images and to emphasize the edges so that the key aspects of the image were emphasized [12,13,14]. Hence, this work aims to study the kinetics of eosinophils during the development of the cellular infiltrate surrounding the nurse cell of *T. spiralis* in experimentally infected mice.

## 2. Results

### 2.1. Larva Development at Muscular Phase

The tongues and diaphragms of experimentally infected mice were collected from day 13 to 39 post infection. There were no deaths due to unexpected causes. Each of the 27 diaphragms was stained with Giemsa to determine the growth of the muscle larva. Figure 1 shows the growth of the larva in each one of the diaphragms. On days 13, 14, and 15, no larvae were observed. From day 16 post infection, the larvae that were invading the muscle began to differentiate. The length of these larvae was 217.97 ± 54.43 µm. Over time, it was observed how the larvae increased in volume and started to roll, while the collagen capsule that surrounds the nurse cell was clearly distinguished in the skeletal muscle. The larval measurement at day 39 post infection was 1,006.08 ± 142.25 µm. Figure 2 shows that larval growth had a linear trend (R^2^ = 0.9185).

### 2.2. Nurse Cell Development

During the course of infection, nurse cells started acquiring a more defined ovoid form (Figure 3). On days 13, 14, and 15, no parasite structures were found; they appeared on day 16 post infection. On day 16, nurse cells had a length of 21.70 ± 7.71 µm along their major axis and 17.26 ± 7.72 µm along the minor axis. On day 39 post infection, the major axis length was 327.60 ± 52.29 µm and minor axis length was 155.52 ± 48.87 µm. Figure 4A shows the linear growth tendency on major (R^2^ = 0.935) and minor (R^2^ = 0.971) axes. Figure 4B shows the area of the nurse cell. On day 16 post infection, the initial area was 334.09 ± 124.43 µm^2^, and on day 39 it was 39,963 ± 8463.36 µm^2^. The calculated linear regression coefficient was R^2^ = 0.8891.

### 2.3. Eosinophil Kinetics

Tissue slides from 27 tongues were stained with erythrosine B to identify the presence of eosinophils surrounding the nurse cells. Eosinophils were mainly distributed on the edge of the nurse cell, so the counting was done in this 10,000 µm^2^ proximal area. No eosinophils were observed on days 13, 14, and 15 post infection; on day 16, bright, red-stained eosinophils were observed (Figure 5). During the infection, the size of the cellular infiltrate increased unexpectedly, and eosinophil kinetics show a multimodal trend, with a bimodal predominance, reaching maximum peaks on days 21 (30 ± 6.22 eosinophils/10,000 µm^2^) and 27 (26 ± 0.71 eosinophils/10,000 µm^2^) post infection. The initial eosinophil count was 13 ± 7.54 cells in 10,000 µm^2^ (Figure 6).

## 3. Discussion

The histological studies of *Trichinella spiralis* infected muscle show nurse cells surrounded by infiltrates resembling a granuloma. Although the infiltrate is composed of macrophages, eosinophils, scattered plasma cells, and lymphoblasts, the proportion of each cellular population is controversial. In immunocompetent hosts, eosinophils are prominent, but in their absence, the muscle larvae die [6,15,16,17,18]. Thus, eosinophils seem to act as immunomodulators to avoid the Th1 response, which is harmful to muscular larvae survival. In this work, we studied the kinetics of eosinophils during the development of the cellular infiltrate surrounding the nurse cell of *T. spiralis* in experimentally infected mice.

Our data show that the ontogenetic development of muscular larvae and nurse cells has exponential growth (Figure 2 and Figure 4). Larvae size went from 217.97 ± 54.43 µm (day 16 post infection) to 1006.08 ± 142.25 µm (day 39 post infection), which can be translated to a growth rate of 34.4 μm/day. Meanwhile, nurse cell development went from (21.70 ± 7.71) × (17.26 ± 7.72) µm (day 16 post infection) to (327.60 ± 52.29) × (155.52 ± 48.87) µm (day 39 post infection), which represents a rate of 13.3 × 6 μm/day. It has been reported that larvae in the female uterus have a length of 100 μm and newborn larvae are 130 μm long, while in the skeletal muscle, larvae increase their size and can measure from 0.65 to 1.45 mm length and 0.026 to 0.040 mm width [19,20].

In the literature, it has been reported that the measurements of the nurse cells are 300–450 μm in length and 150–300 μm in width [21]. To the best of our knowledge, this is the first report about nurse cell exponential growth correlated to the larvae size increment during their development. This fact is expected because larvae need space to grow per their metabolic needs. The nurse cell is a specialized structure for harboring *Trichinella spiralis* larvae, providing the nutrition required for rapid growth (about a 40% increase in volume per day) and removing any unnecessary metabolic-waste products [5]. In general, the relationship of one larva by each nurse cell is expected; however, there are reports of two or more larvae per nurse cell, without prejudice of the morphophysiology of each larva [9]. Since the report of Perez and Luengo in 1969 [22], where nine muscular larvae inside of a single nurse cell were detected in a muscle biopsy from a massively infected pig, it has been suggested that this phenomenon could be related to the intensity of infection, i.e., more than one stage 1 larva invades the myocyte at nearly the same time. There is no information regarding the possible fusion of two or more nurse cells during the infection process; the cellular exponential growth occurs according to the development of the parasite.

The nurse cell parasite complex is surrounded by a collagen capsule and consists predominantly of two collagen types, IV and VI, both of which are synthesized by the nurse cell [3]. The capsule wall is a prominent non-cellular structure that has two layers. The inner is produced by the nurse cell, and the outer is produced by fibroblasts around the capsule. It has been hypothesized that the ES proteins produced by the larva act as a messenger for communication between the parasite and muscle cells, which allows for long-term coexistence. Several hypotheses have been exposed to explain the collagen cyst biological function, such as larvae protection against freezing, nutrient supply, or protection from the immune system. In this study, we show that in tissue samples, the first cells of the cellular infiltrate are observed simultaneously to the first collagen capsules (Figure 3); thus, according to the observations of Sacchi et al., in 2001 [23], the thickness of the capsule could be associated with the chronicity of the parasitosis; thick capsules (22.7 to 33.5 μm) of horses suggest old infections, while thin capsules (average 18.7 μm) suggest recent infections. Although we did not measure capsule growth kinetics, our data also suggest that capsule thickness is associated with nurse cell growth; for example, the thickness of the capsules at day 20 post infection was on average 10 μm, while at day 39 it was 50 μm.

Although the cellular lineages’ kinetics in the infiltrate are controversial, there are some reports of macrophages, eosinophils, basophils, mast cells, and T and B lymphocytes. Indeed, Wu et al. in 2008 [4], described the nurse cell formation with emphasis on an analogy to muscle cell repair, where early events are an accumulation of inflammatory cells in the damaged site. The activated mononuclear cells release factors that provide chemotactic signals to other inflammatory cells; thus, the neutrophils and macrophages must be the first to come. Our data, obtained from the murine model, suggest that during the formation of the cellular infiltrate, eosinophils are not the main component, nor are their numbers constant. However, other authors have suggested that the cellular infiltrate mainly consists of eosinophils [15].

In this work, we observe that eosinophils are recruited around the nurse cell from day 16 after infection. However, recruitment is not continuous but rather follows a predominantly bimodal trend, with peaks on days 21 and 27 after infection (Figure 6). A possible explanation for this kinetic behavior is that eosinophils, upon reaching their destination, could degranulate and lose arginine, which is one of the main components of the basic protein of eosinophils and that selectively stains with erythrosine B [24]. In this way, in the absence of arginine, the eosinophils would not be visible. Alternatively, a second explanation is that the immune response favored by eosinophils is more beneficial for the parasite than for the host. It has previously been documented that the survival and growth of stage 1 larvae depend on the presence of eosinophils that are recruited from skeletal muscle [6,15,16,25,26]. Due to the size and motility of the parasites, tissue damage warrants regenerative intervention [4]. In this way, the production of IL-4 by eosinophils would contribute to the repair of muscle damage caused by stage 1 larvae [26]. Additionally, several authors have published reports describing that during the formation of nurse cells, several cell types are recruited, among them eosinophils and CD4 cells, which could control the inflammation caused by the parasite 40 days after infection [11,15,27].

In conclusion, our data suggest that the presence of eosinophils is essential at two crucial moments in the life cycle of the parasite. First, when the newly hatched larvae invade the myocyte and cause damage to the myocyte. This action triggers a muscle regeneration mechanism that could involve the presence of eosinophils. Second, eosinophils, by regulating the host’s immune response, ensure the survival of the larvae. Further studies on the molecular and cellular interaction in the host–parasite relationship will provide a unique insight into the immunobiology of the parasite that could help to understand the inflammatory mechanisms involved during parasitosis. It is worth mentioning that the sample size of the experimentally infected mice was small. That is important for the interpretation of our findings, and therefore in the future it would be convenient to examine a greater number of samples to corroborate the results we obtained.

## 4. Material and Methods

### 4.1. Animals and Infection

A total of sixty male CD1 mice of 6 weeks of age and 20 ± 1 g of weight were experimentally infected by oral administration with 50 muscular larvae of *Trichinella spiralis* MSUS/MEX/91/CM-91. The experimental infection was equivalent to two muscular larvae per gram of body weight (ML/g). The muscle larvae were recovered by conventional pepsin-HCl artificial digestion of minced carcasses from previously infected mice [28]. All animals were housed in controlled light and temperature conditions and handled following the Official Mexican Standard NOM-062-ZOO-1999 for the care and use of laboratory animals. The protocol was approved on 7 December 2018, with code 010/CIECUA/1, by the Ethics and Animal Care Committee-ICAP, Autonomous University of the State of Hidalgo; the research protocol was approved on 21 June 2019, with the code INVI-045-2019 by the Immunological Research Coordination, Institute for Epidemiological Diagnosis and Reference (InDRE). To characterize the cellular infiltrate during development of the nurse cell, the tongue and diaphragm were taken daily from the carcasses of the experimentally infected mice. Two mice were slaughtered and processed from day 13 post infection and finished at day 39. The tongues were subjected to histological sections to characterize the development of the cellular infiltrate and the eosinophil kinetics. The diaphragms were stained with Giemsa to measure the development of the nurse cell and the muscular larvae.

### 4.2. Giemsa Staining

The diaphragms were cut into 3–4 mm pieces and compressed between two glass slides. Giemsa staining was carried out as previously reported [9,10]. Immediately, the tissue samples were compressed in a formaldehyde-acetic alcohol solution for 4 h and then transferred to a 50% ethyl alcohol solution. The samples were stained in 10 mL of a Giemsa 1:6 solution in 0.01 M phosphates, pH 7.2 for 45 min at room temperature with slow constant stirring. The samples were then transferred to acidic alcohol (0.02 N HCl in 50% ethyl alcohol) for 45 s and dehydrated for 2 to 3 min in alcohol (30%, 50%, 70%, and 100%) with gentle shaking. The samples were then incubated in a mixture of absolute ethyl alcohol and xylene and finally in absolute xylene for permanent mounting.

### 4.3. Histological Sections and Staining Methods

Tongue samples (0.3 cm^3^) were immersed in paraffin, treated with conventional dehydration, inclusion, and staining methods. Tongues were cut into 7 µm-long pieces and stained with hematoxylin-eosin or erythrosine B. To analyze the cellular infiltrate, the histological sections were stained with hematoxylin dye for 15 min and with eosin for 5 min by a conventional method. To determine eosinophil presence, tongues were placed in glass slides and stained with Harris’s hematoxylin (1:3) for 7 min, using 0.3 to 0.4 mL for each sample. Samples were then rinsed using tap water until a color change was observed and rinsed afterward with distilled water. Each glass slide was stained with 0.015% erythrosine B in a 0.1 M glycine buffer solution, pH 10 for 30 min. Next, the glass slides were treated with 70% ethanol [21].

### 4.4. Image Analysis

Image-Pro-Plus^®^ was used for image analysis (Media Cybernetics Inc., Rockville, MD, USA). Images were changed to greyscale (8-bit) using the ImageJ software (National Institutes of Health, Bethesda, MD, USA). Segmentation was carried out using the Otsu algorithm [29]. The morphometric parameters of the muscle larvae and the nurse cell were measured using the same software. The eosinophil kinetics in the cellular infiltrate were also evaluated. Micrographs were analyzed by counting the occurrence of eosinophils around the nurse cell in an area of 10,000 μm^2^. Each parameter was determined by the measurement of 10 larvae/nurse cells.

### 4.5. Statistical Analysis

Values of the measured parameters were expressed as mean ± standard deviation. Statistical analysis was performed by linear regression and *t*-test using GraphPad Prism software (GraphPad Software, La Jolla, CA, USA). The difference was considered significant at *p*-value < 0.05.

## Figures and Tables

**Figure 1 pathogens-10-01382-f001:**
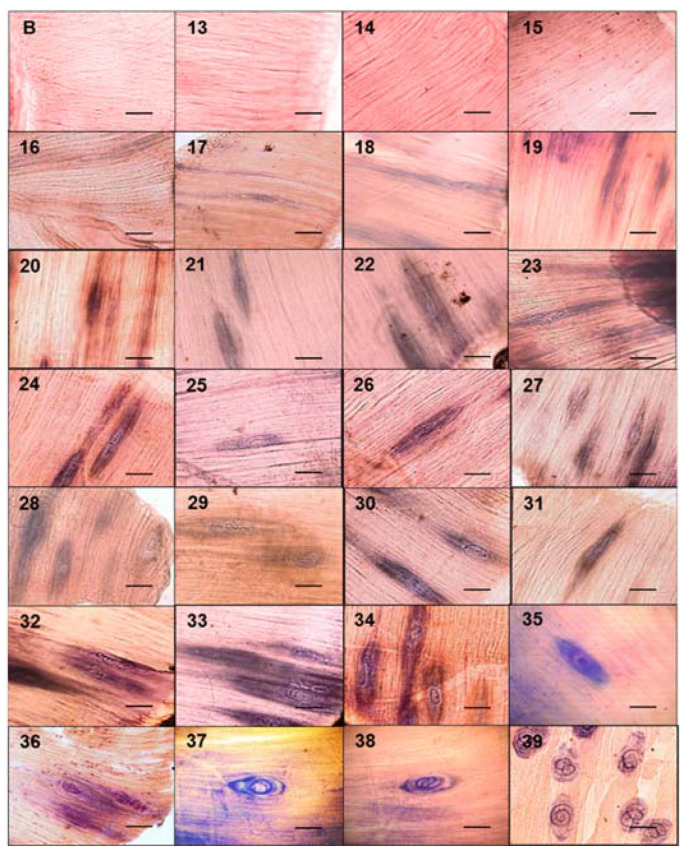
Development of the nurse cell and muscle larvae complex in mice experimentally infected with *Trichinella spiralis.* The diaphragms of the experimentally infected mice were taken and stained with Giemsa. They were then observed under an optical microscope at 20×.The muscular larvae were stained blue. The 50 μm bar (bottom right corner) and the day post infection when the sample was taken are indicated in each photograph (top left number).

**Figure 2 pathogens-10-01382-f002:**
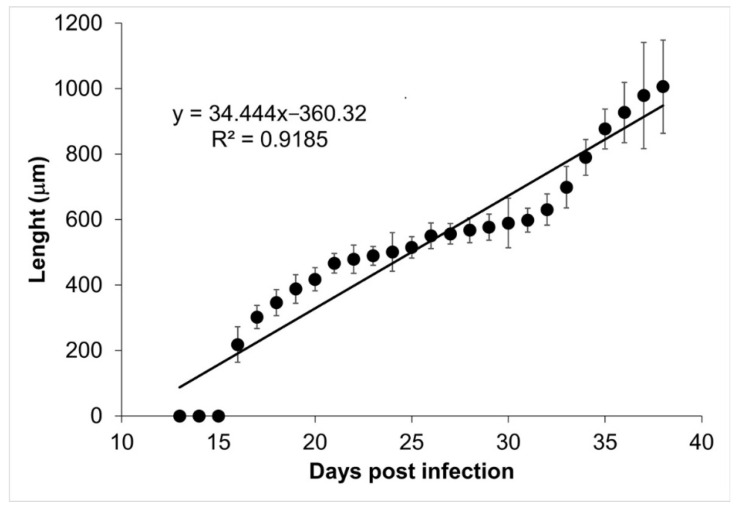
*Trichinella spiralis* muscle larvae longitudinal development. The length of the muscular larvae is shown as a function of the day post infection. The linear regression function is shown at the top left corner of the figure.

**Figure 3 pathogens-10-01382-f003:**
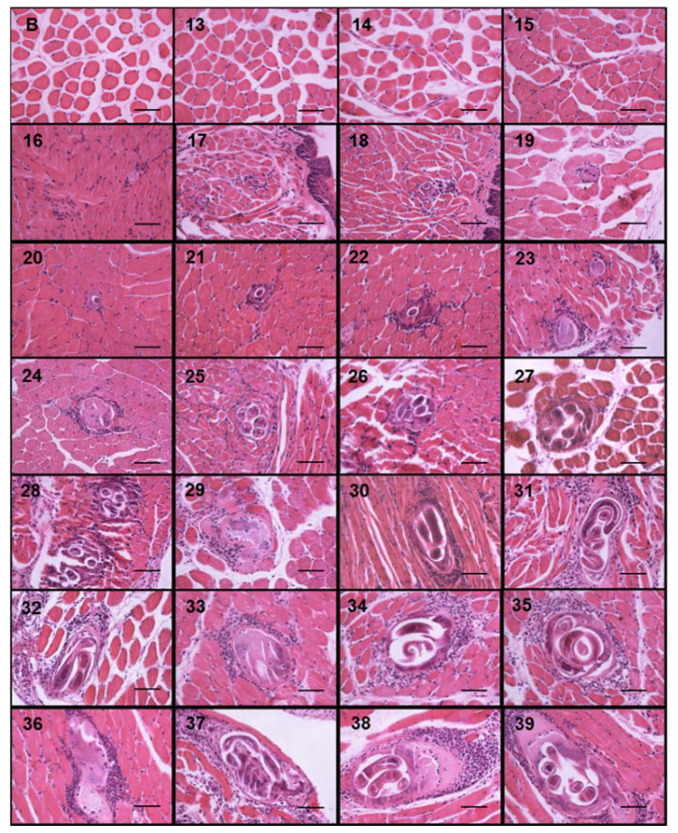
*Trichinella spiralis* nurse cell and cellular infiltrate development. Histological sections of the tongues of experimentally infected mice were collected from day 13 to 39 post infection and stained with hematoxylin and eosin. Photographs were taken with a light microscope at 40×. The nurse cell was stained purple, the cellular infiltrate strong purple, and the muscle tissue pink. The 50 μm bar (bottom right corner) and the day post infection when the sample was taken are indicated in each photograph (top left number) post infection.

**Figure 4 pathogens-10-01382-f004:**
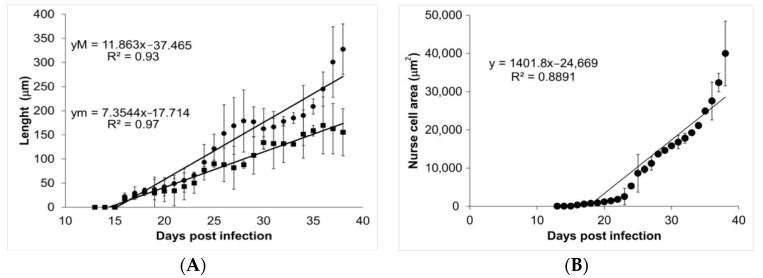
Analysis of nurse cell development. Panel (**A**) shows the length of the major (closed circles) and minor (closed squares) axes. The values of the linear regression function for the major (yM) and minor (ym) axes are shown in the upper left corner. Panel (**B**) shows the area of the nurse cell as a function of the days post infection. The value of the linear regression function is shown in the top left corner of the panel.

**Figure 5 pathogens-10-01382-f005:**
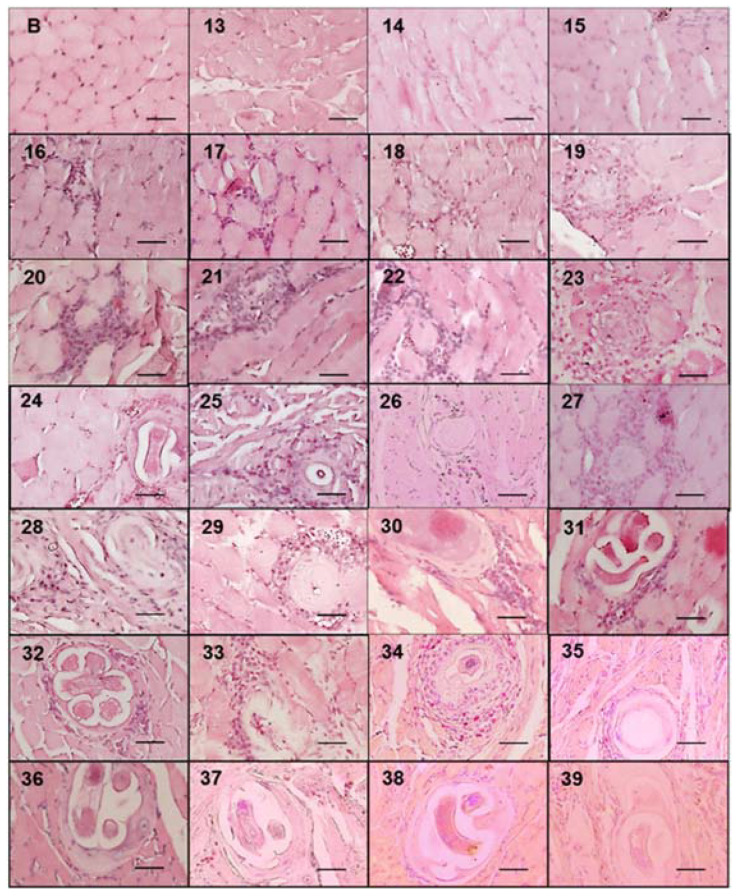
Eosinophils that make up the cell infiltrate surrounding the nurse cell. The figure shows images of histological sections from experimentally infected mice tongues collected from day 13 to 39 post infection and stained with erythrosine B. Photographs were taken with a light microscope at 40×. The appearance of eosinophils (red stained) is shown around the nurse cell. The 50 μm bar (bottom right corner) and the day post infection when the sample was taken are indicated in each photograph (top left number) post infection.

**Figure 6 pathogens-10-01382-f006:**
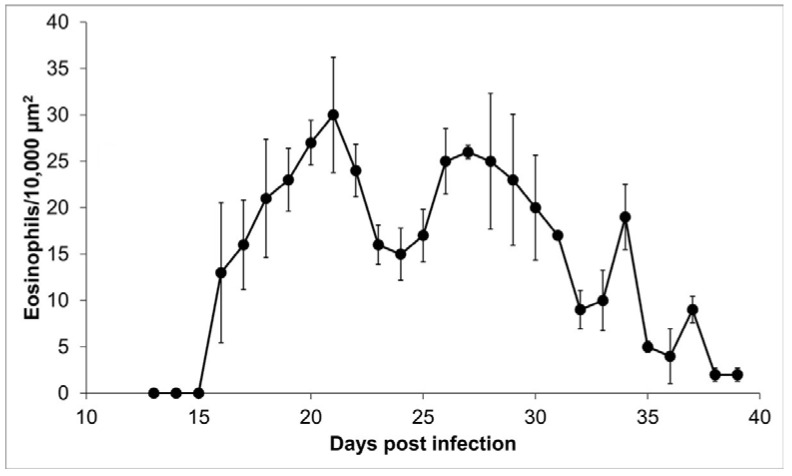
Eosinophils within the cellular infiltrate around the nurse cell. Presence of eosinophils in 10,000 μm^2^ of the area around the nurse cell as a function of day post infection.

## Data Availability

Data supporting the conclusions of this article are included within the article and its additional file. The datasets used and/or analyzed during the present study are available from the corresponding author upon reasonable request.

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
