# Peer review of "Kinetics of Eosinophils during Development of the Cellular Infiltrate Surrounding the Nurse Cell of Trichinella spiralis in Experimentally Infected Mice"

_pathogens, 2021, doi:10.3390/pathogens10111382_

Round 1
Reviewer 1 Report
Despite this study contribute to understanding cellular infiltrate, it is important to emphasize that is in murine model.
Author Response
Despite this study contribute to understanding cellular infiltrate, it is important to emphasize that is in murine model.
Response: The authors appreciate the reviewer's time. In the abstract (line 18), introduction (line 93) and discussion (line 263) sections it is emphasized that the results obtained were in the murine model. In addition, the information from the ethics and research committees was added in the section 4.1 “Animals and infection”. Please see the line 301: “All animals were housed in controlled light and temperature conditions and handled following the Mexican Regulation NOM-062-ZOO-1999 for the care and use of laboratory animals. The protocol was approved on December 7, 2018 with code 010/ CIECUA/1, by the Ethics and Animal Care Committee-ICAP, Autonomous University of the State of Hidalgo; the research protocol was approved on June 21, 2019 with the code INVI-045-2019 by Immunological Research Coordination, Institute for Epidemiological Diagnosis and Reference”.
Reviewer 2 Report
In my opinion, this is a well-structured manuscript with very nice photos. Nevertheless, I have a few suggestions for the authors:
- English language should be improved in the text. I give a few examples.
Line 32: I believe you should write Trichinella spiralis instead of Trichinella and add a keyword that will state that you performed experimental infection.
Line 43: I think it’s better to delete the world ‘a’ before ‘nurse cells’.
Line 86: You need to rephrase the sentence.
Lines 93-94: You should change some verbs to past tense.
Line 100, 118, 138, 139: You should decide if you want to use present or past tense.
Line 144: You should correct Trichinella to Trichinella spiralis.
Lines 148, 150: The word ‘thus’ is better to be replaced in the first or the second sentence.
Lines 201-202: I believe you should rephrase the sentence to make it more clearly presented.
Line 221: The word ‘the’ in the phrase ‘the first’ should be omitted.
- More references should be added.
Line 145 - 147: You should add references.
Lines 219: You need to add more references, since you refer to several authors.
- You should state clearly the number of experimentally infected mice in the section of Materials and Methods.
Line 230: You should state the number of experimentally infected mice.
Line 240: You should state the number of the examined mice per day.
- In the Discussion section, you should state that the sample size of the experimentally infected mice was small. That’s important for the interpretation of your findings and the future perspectives of your work.
Author Response
ITEM 1
In my opinion, this is a well-structured manuscript with very nice photos. Nevertheless, I have a few suggestions for the authors:
English language should be improved in the text. I give a few examples.
Line 32: I believe you should write Trichinella spiralis instead of Trichinella and add a keyword that will state that you performed experimental infection.
Line 43: I think it’s better to delete the world ‘a’ before ‘nurse cells’.
Line 86: You need to rephrase the sentence.
Lines 93-94: You should change some verbs to past tense.
Line 100, 118, 138, 139: You should decide if you want to use present or past tense.
Line 144: You should correct Trichinella to Trichinella spiralis.
Lines 148, 150: The word ‘thus’ is better to be replaced in the first or the second sentence.
Lines 201-202: I believe you should rephrase the sentence to make it more clearly presented.
Line 221: The word ‘the’ in the phrase ‘the first’ should be omitted.
Response ITEM 1: The final version of the manuscript underwent a grammatical and syntax review of the English language for better reading comprehension. The examples provided by the reviewer were corrected according to his suggestion and what was indicated by the grammar review process According the critical review of the English language “Kinetics of eosinophils during development of the cellular infiltrate surrounding the nurse cell of Trichinella spiralis in experimentally infected mice”
ITEM 2
More references should be added.
Line 145 - 147: You should add references.
Response ITEM2: In this paragraph, three references previously cited in the text were rearranged and two new citations were added. Finally, the numbering of the references was re-ordered in the “References” section
references previously cited
6- Fabre V.; Beiting, D. P.; Bliss, S. K.; Gebreselassie N. G.; Gagliardo L. F.; Lee N. A.; Lee J. J.; Appleton J. A. Eo-sinophil deficiency compromises parasite survival in chronic nematode infection. J. Immunol. 2009, 182, 1577–1583. [DOI: 10.4049/jimmunol.182.3.1577]
20 -Fabre, M.V.; Beiting, D.P.; Bliss, S.K.; Appleton, J.A. 2009. Immunity to Trichinella spiralis muscle infection. Vet Parasitol. 159(3-4):245-8. [DOI:10.1016/j.vetpar.2008.10.051].
24- Gebreselassie, N.G.; Moorhead, A.R., Fabre, V.; Gagliardo L.F.; Lee, N.A.; Lee J.J.; Appleton, J.A. Eosinophils preserve parasitic nematode larvae by regulating local immunity. J. Immunol., 2012, 188, 417–425. [DOI: 10.4049/jimmunol.1101980]
new citations
Huang, L.; Beiting, D.P.; Gebreselassie, N. G.; Gagliardo, L.F.; Ruyechan, M.C.; Lee, N.A.; Lee, J.J.; Appleton, J.A. Eosinophils and IL-4 Support Nematode Growth Coincident with an Innate Response to Tissue Injury. PLoS Pathog., 2015, 11, e1005347. [DOI: 10.1371/journal.ppat.1005347]
Mitre, E.; Klion, A.D. Eosinophils and helminth infection: protective or pathogenic?. Semin Immunopathol., 2021, 43, 363–381. [DOI: 10.1007/s00281-021-00870-z]
ITEM 3
Lines 219: You need to add more references, since you refer to several authors.
Response ITEM 3: Two reference previously cited in the text were rearranged and one new citation were added. Finally, the numbering of the references was re-ordered in the “References” section
references previously cited
11-Beiting, D.P.; Gagliardo, L.F.; Hesse, M.; Bliss, S.K.; Meskill, D.; Appleton JA. Coordinated control of immunity to muscle stage Trichinella spiralis by IL-10, regulatory T cells, and TGF-beta. J. Immunol. 2007, 15, 1039–1047. [DOI: 10.4049/jimmunol.178.2.1039]
20 -Fabre, M.V.; Beiting, D.P.; Bliss, S.K.; Appleton, J.A. 2009a. Immunity to Trichinella spiralis muscle infection. Vet Parasitol. 159(3-4):245-8. [DOI:10.1016/j.vetpar.2008.10.051].
new citation
Bruschi, F., Chiumiento, L. Trichinella inflammatory myopathy: host or parasite strategy?. Parasites Vectors., 2011, 4, 42. [DOI: 10.1186/1756-3305-4-42]
ITEM 4
You should state clearly the number of experimentally infected mice in the section of Materials and Methods.
Line 230: You should state the number of experimentally infected mice.
Line 240: You should state the number of the examined mice per day.
Response ITEM 4: In the materials and methods section it is indicated that we infected a total of 60 mice and we examined 2 mice daily.
ITEM 5
In the Discussion section, you should state that the sample size of the experimentally infected mice was small. That’s important for the interpretation of your findings and the future perspectives of your work.
Response ITEM 5: In the discussion section, we state clearly state that the size of the experimentally infected mice was small. Please see line 290, where the next text were added: “It is worth mentioning that the sample size of the experimentally infected mice was small. That’s important for the interpretation of our findings and therefore in the future it would be convenient to examine a greater number of samples to corroborate the results we obtained”
Reviewer 3 Report
English needs attention, preferably by a native speaker of the language, as many incongruities could be detected regarding syntax (e.g. singular vs. plural, etc.).
In addition, the style of writing would also benefit from a review of verbal tenses. For example, the abstract would read better as “We studied…” instead of “Here we study…”.
Line 20 – consider writing to read as: Tongues and diaphragms were obtained daily from days 13 to 39 post-infection
Line 27 – “newborn” larvae is not an adequate designation – replace with stage 1 larvae or L1 larvae – change accordingly throughout the manuscript
Abbreviate Trichinella spirallis as T. spiralis after its first use, both in the abstraction and main text.
Keywords – display alphabetically
Line 38 – rewrite to read as: two different TYPES of clinical manifestations
“Parental stage” does not seem to be an adequate designation. In fact, parental stage would refer to adult stage, which is not present in muscles.
Not clear: … and, subsequent the nurse cell formation in the muscle
Line 43: a “nurse cells” OR “nurse cells” [without “a”] – singular or plural?
Line 59 – explain the meaning of “iNos”
Line 86 – the meaning of Larva Development Muscular is not quite clear
Line 276 – please mention which statistical tests were used.
References – titles should be presented with lowercase or uppercase – do not mix both styles
Author Response
ITEM 1
English needs attention, preferably by a native speaker of the language, as many incongruities could be detected regarding syntax (e.g. singular vs. plural, etc.).
In addition, the style of writing would also benefit from a review of verbal tenses. For example, the abstract would read better as “We studied…” instead of “Here we study…”.
Response ITEM 1: The authors appreciate the time of the reviewer and, according to their comment, the article was sent to a process to review the graphics and syntax of the English language. Please see the acknowledgements. Indeed, the paper is now entitled, according the critical review of the English language as “Kinetics of eosinophils during development of the cellular infiltrate surrounding the nurse cell of Trichinella spiralis in experimentally infected mice”
ITEM 2
Line 20 – consider writing to read as: Tongues and diaphragms were obtained daily from days 13 to 39 post-infection
Response ITEM 2: the change in sentence was made, please see line 20
ITEM 3
Line 27 – “newborn” larvae is not an adequate designation – replace with stage 1 larvae or L1 larvae – change accordingly throughout the manuscript
Response ITEM 3: the change in sentence was made along the text. Stage 1 larvae was used instead of newborn larvae
ITEM 4
Abbreviate Trichinella spirallis as T. spiralis after its first use, both in the abstraction and main text.
Response ITEM 4: the change in sentence was made along the text.
ITEM 5
Keywords – display alphabetically
Response ITEM 5: the change in Keywords was made, please see line 32
ITEM 6
Line 38 – rewrite to read as: two different TYPES of clinical manifestations
Response ITEM 6: the change in sentence was made
ITEM 7
“Parental stage” does not seem to be an adequate designation. In fact, parental stage would refer to adult stage, which is not present in muscles.
Not clear: … and, subsequent the nurse cell formation in the muscle
Line 43: a “nurse cells” OR “nurse cells” [without “a”] – singular or plural?
Response ITEM 7: For a better understanding of the text, "parental stage" was changed to "muscular phase" and it was rewritten in the text: "The muscular phase begins with the infection of a myofibrilla with the stage 1 larvae, which will develop into a muscular larva". Indeed, “parental stage” was changed along the text
ITEM 8
Line 59 – explain the meaning of “iNos”
Response ITEM 8: iNos was changed by inducible nitric oxide synthase, please see line 74
ITEM 9
Line 86 – the meaning of Larva Development Muscular is not quite clear
Response ITEM 9: The title was rewritten by "Larva Development at Muscular Phase". Please see line 102
ITEM 10
Line 276 – please mention which statistical tests were used.
Response ITEM 10: in line 396, the word “t-test” was added.
ITEM 11
References – titles should be presented with lowercase or uppercase – do not mix both styles
Response ITEM 11: Each reference was reviewed and corrected with lower case.